# Differentiable Learning-to-Normalize via Switchable Normalization

**Ping Luo**[1,3*]  **Jiamin Ren**[2*]  **Zhanglin Peng**[2]  **Ruimao Zhang**[1]  **Jingyu Li**[1]
[1]The Chinese University of Hong Kong    [2]SenseTime Research    [3]The University of Hong Kong

## Abstract

We address a learning-to-normalize problem by proposing Switchable Normalization (SN), which learns to select different normalizers for different normalization layers of a deep neural network. SN employs three distinct scopes to compute statistics (means and variances) including a channel, a layer, and a minibatch. SN switches between them by learning their importance weights in an end-to-end manner. It has several good properties. First, it adapts to various network architectures and tasks (see Fig.1). Second, it is robust to a wide range of batch sizes, maintaining high performance even when small minibatch is presented (*e.g.* 2 images/GPU). Third, SN does not have sensitive hyper-parameter, unlike group normalization that searches the number of groups as a hyper-parameter. Without bells and whistles, SN outperforms its counterparts on various challenging benchmarks, such as ImageNet, COCO, CityScapes, ADE20K, and Kinetics. Analyses of SN are also presented. We hope SN will help ease the usage and understand the normalization techniques in deep learning. The code of SN has been released in `https://github.com/switchablenorms/`.

## 1 Introduction

Normalization techniques are effective components in deep learning, advancing many research fields such as natural language processing, computer vision, and machine learning. In recent years, many normalization methods such as Batch Normalization (BN) (Ioffe & Szegedy, 2015), Instance Normalization (IN) (Ulyanov et al., 2016), and Layer Normalization (LN) (Ba et al., 2016) have been developed. Despite their great successes, existing practices often employed the same normalizer in all normalization layers of an entire network, rendering suboptimal performance. Also, different normalizers are used to solve different tasks, making model design cumbersome.

To address the above issues, we propose *Switchable Normalization (SN)*, which combines three types of statistics estimated channel-wise, layer-wise, and minibatch-wise by using IN, LN, and BN respectively. SN switches among them by learning their importance weights. By design, *SN is adaptable to various deep networks and tasks*. For example, the ratios of IN, LN, and BN in SN are compared in multiple tasks as shown in Fig.1 (a). We see that using one normalization method uniformly is not optimal for these tasks. For instance, image classification and object detection prefer the combination of three normalizers. In particular, SN chooses BN more than IN and LN in image classification and the backbone network of object detection, while LN has larger weights in the box and mask heads. For artistic image style transfer (Johnson et al., 2016), SN selects IN. For neural architecture search, SN is applied to LSTM where LN is preferable than group normalization (GN) (Wu & He, 2018), which is a variant of IN by dividing channels into groups.

The selectivity of normalizers makes *SN robust to minibatch size*. As shown in Fig.1 (b), when training ResNet50 (He et al., 2016) on ImageNet (Deng et al., 2009) with different batch sizes, SN is close to the "ideal case" more than BN and GN. For $(8, 32)$ as an example[1], ResNet50 trained with SN is able to achieve 76.9% top-1 accuracy, surpassing BN and GN by 0.5% and 1.0% respectively. In general, SN obtains better or comparable results than both BN and GN in all batch settings.

---

*The first two authors contribute equally. Corresponding to: pluo.lhi@gmail.com, {renjiamin, pengzhanglin, zhangruimao, lijingyu}@sensetime.com.

[1]In this work, minibatch size refers to the number of samples per GPU, and batch size is '#GPUs' times '#samples per GPU'. A batch setting is denoted as a 2-tuple, *(#GPUs, #samples per GPU)*.

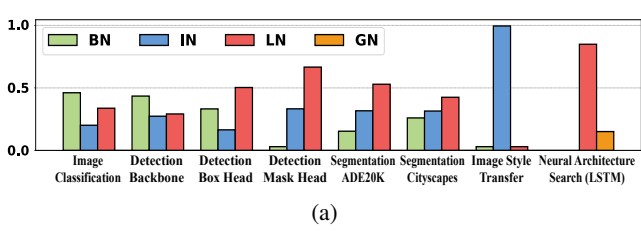
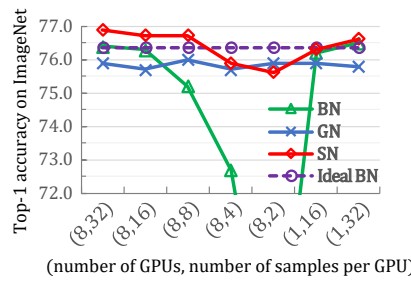

(a)

(number of GPUs, number of samples per GPU)

(b)

Figure 1: (a) shows that SN adapts to various networks and tasks by learning importance ratios to select normalizers. In (a), a ratio is between 0 and 1 and all ratios of each task sum to 1. (b) shows the top-1 accuracies of ResNet50 trained with SN on ImageNet and compared with BN and GN in different batch settings. The gradients in training are averaged over all GPUs and the statistics of normalizers are estimated in each GPU. For instance, all methods are compared to an ideal case, 'ideal BN', whose accuracies are 76.4% for all settings. This ideal case cannot be obtained in practice. In fact, when the minibatch size decreases, BN's accuracies drop significantly, while SN and GN both maintain reasonably good performance. SN surpasses or is comparable to both BN and GN in all settings.

Overall, this work has three key **contributions**. (1) We introduce Switchable Normalization (S-N), which is applicable in both CNNs and RNNs/LSTMs, and improves the other normalization techniques on many challenging benchmarks and tasks including image recognition in ImageNet (Russakovsky et al., 2015), object detection in COCO (Lin et al., 2014), scene parsing in Cityscapes (Cordts et al., 2016) and ADE20K (Zhou et al., 2017), artistic image stylization (Johnson et al., 2016), neural architecture search (Pham et al., 2018), and video recognition in Kinetics (Kay et al., 2017). (2) The analyses of SN are presented where multiple normalizers can be compared and understood with geometric interpretation. (3) By enabling *each normalization layer in a deep network to have its own operation*, SN helps ease the usage of normalizers, pushes the frontier of normalization in deep learning, as well as opens up new research direction. We believe that all existing models could be reexamined with this new perspective. We'll make the code of SN available and recommend it as an alternative of existing handcrafted approaches.

In the following sections, we first present SN in Sec.2 and then discuss its relationships with previous work in Sec.3. SN is evaluated extensively in Sec.4.

## 2 SWITCHABLE NORMALIZATION (SN)

We describe a general formulation of a normalization layer and then present SN.

**A General Form.** We take CNN as an illustrative example. Let $h$ be the input data of an arbitrary normalization layer represented by a 4D tensor $(N, C, H, W)$, indicating number of samples, number of channels, height and width of a channel respectively as shown in Fig.2. Let $h_{ncij}$ and $\hat{h}_{ncij}$ be a pixel before and after normalization, where $n \in [1, N]$, $c \in [1, C]$, $i \in [1, H]$, and $j \in [1, W]$. Let $\mu$ and $\sigma$ be a mean and a standard deviation. We have

$$\hat{h}_{ncij} = \gamma \frac{h_{ncij} - \mu}{\sqrt{\sigma^2 + \epsilon}} + \beta, \qquad (1)$$

where $\gamma$ and $\beta$ are a scale and a shift parameter respectively. $\epsilon$ is a small constant to preserve numerical stability. Eqn.(1) shows that each pixel is normalized by using $\mu$ and $\sigma$, and then re-scale and re-shift by $\gamma$ and $\beta$.

IN, LN, and BN share the formulation of Eqn.(1), but they use different sets of pixels to estimate $\mu$ and $\sigma$. In other words, the num-

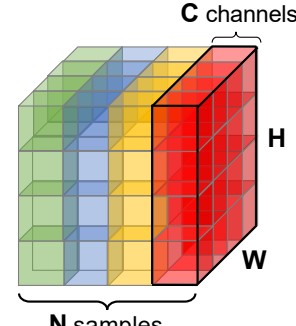

Figure 2: The size of feature maps is $N \times C \times H \times W$ ($N = 4$ in this example). Different normalizers estimate statistics along different axes.

bers of their estimated statistics are different. In general, we have

$$\mu_k = \frac{1}{|I_k|} \sum_{(n,c,i,j) \in I_k} h_{ncij}, \quad \sigma_k^2 = \frac{1}{|I_k|} \sum_{(n,c,i,j) \in I_k} (h_{ncij} - \mu_k)^2, \tag{2}$$

where $k \in \{\text{in}, \text{ln}, \text{bn}\}$ is used to distinguish different methods. $I_k$ is a set pixels and $|I_k|$ denotes the number of pixels. Specifically, $I_{\text{in}}$, $I_{\text{ln}}$, and $I_{\text{bn}}$ are the sets of pixels used to compute statistics in different approaches.

IN was established in the task of artistic image style transfer (Johnson et al., 2016; Huang & Belongie, 2017). In IN, we have $\mu_{\text{in}}, \sigma_{\text{in}}^2 \in \mathbb{R}^{N \times C}$ and $I_{\text{in}} = \{(i,j)|i \in [1, H], j \in [1, W]\}$, meaning that IN has $2NC$ elements of statistics, where each mean and variance value is computed along $(H, W)$ for each channel of each sample.

LN (Ba et al., 2016) was proposed to ease optimization of recurrent neural networks (RNNs). In LN, we have $\mu_{\text{ln}}, \sigma_{\text{ln}}^2 \in \mathbb{R}^{N \times 1}$ and $I_{\text{ln}} = \{(c,i,j)|c \in [1, C], i \in [1, H], j \in [1, W]\}$, implying that LN has $2N$ statistical values, where a mean value and a variance value are computed in $(C, H, W)$ for each one of the $N$ samples.

BN (Ioffe & Szegedy, 2015) was first demonstrated in the task of image classification (He et al., 2016; Krizhevsky et al., 2012) by normalizing the hidden feature maps of CNNs. In BN, we have $\mu_{\text{bn}}, \sigma_{\text{bn}}^2 \in \mathbb{R}^{C \times 1}$ and $I_{\text{bn}} = \{(n,i,j)|n \in [1, N], i \in [1, H], j \in [1, W]\}$, in the sense that BN treats each channel independently like IN, but not only normalizes across $(H, W)$, but also the $N$ samples in a minibatch, leading to $2C$ elements of statistics.

## 2.1 FORMULATION OF SN

SN has an intuitive expression

$$\hat{h}_{ncij} = \gamma \frac{h_{ncij} - \Sigma_{k \in \Omega} w_k \mu_k}{\sqrt{\Sigma_{k \in \Omega} w_k' \sigma_k^2 + \epsilon}} + \beta, \tag{3}$$

where $\Omega$ is a set of statistics estimated in different ways. In this work, we define $\Omega = \{\text{in}, \text{ln}, \text{bn}\}$ the same as above where $\mu_k$ and $\sigma_k^2$ can be calculated by following Eqn.(2). However, this strategy leads to large redundant computations. In fact, the three kinds of statistics of SN depend on each other. Therefore we could reduce redundancy by reusing computations,

$$\mu_{\text{in}} = \frac{1}{HW} \sum_{i,j}^{H,W} h_{ncij}, \quad \sigma_{\text{in}}^2 = \frac{1}{HW} \sum_{i,j}^{H,W} (h_{ncij} - \mu_{\text{in}})^2,$$

$$\mu_{\text{ln}} = \frac{1}{C} \sum_{c=1}^{C} \mu_{\text{in}}, \quad \sigma_{\text{ln}}^2 = \frac{1}{C} \sum_{c=1}^{C} (\sigma_{\text{in}}^2 + \mu_{\text{in}}^2) - \mu_{\text{ln}}^2,$$

$$\mu_{\text{bn}} = \frac{1}{N} \sum_{n=1}^{N} \mu_{\text{in}}, \quad \sigma_{\text{bn}}^2 = \frac{1}{N} \sum_{n=1}^{N} (\sigma_{\text{in}}^2 + \mu_{\text{in}}^2) - \mu_{\text{bn}}^2, \tag{4}$$

showing that the means and variances of LN and BN can be computed based on IN. Using Eqn.(4), the computational complexity of SN is $\mathcal{O}(NCHW)$, which is comparable to previous work.

Furthermore, $w_k$ and $w_k'$ in Eqn.(3) are importance ratios used to weighted average the means and variances respectively. Each $w_k$ or $w_k'$ is a scalar variable, which is shared across all channels. There are $3 \times 2 = 6$ importance weights in SN. We have $\Sigma_{k \in \Omega} w_k = 1, \Sigma_{k \in \Omega} w_k' = 1$, and $\forall w_k, w_k' \in [0, 1]$, and define

$$w_k = \frac{e^{\lambda_k}}{\Sigma_{z \in \{\text{in}, \text{ln}, \text{bn}\}} e^{\lambda_z}} \quad \text{and} \quad k \in \{\text{in}, \text{ln}, \text{bn}\}. \tag{5}$$

Here each $w_k$ is computed by using a softmax function with $\lambda_{\text{in}}$, $\lambda_{\text{ln}}$, and $\lambda_{\text{bn}}$ as the control parameters, which can be learned by back-propagation (BP). $w_k'$ are defined similarly by using another three control parameters $\lambda_{\text{in}}'$, $\lambda_{\text{ln}}'$, and $\lambda_{\text{bn}}'$.

**Training.** Let $\Theta$ be a set of network parameters (*e.g.* filters) and $\Phi$ be a set of control parameters that control the network architecture. In SN, we have $\Phi = \{\lambda_{\text{in}}, \lambda_{\text{ln}}, \lambda_{\text{bn}}, \lambda_{\text{in}}', \lambda_{\text{ln}}', \lambda_{\text{bn}}'\}$. Training

a deep network with SN is to minimize a loss function $\mathcal{L}(\Theta, \Phi)$, where $\Theta$ and $\Phi$ can be optimized jointly by back-propagation (BP). This training procedure is different from previous meta-learning algorithms such as network architecture search (Colson et al., 2007; Liu et al., 2018; Pham et al., 2018). In previous work, $\Phi$ represents as a set of network modules with different learning capacities, where $\Theta$ and $\Phi$ were optimized in two BP stages iteratively by using two training sets that are non-overlapped. For example, previous work divided an entire training set into a training and a validation set. However, if $\Theta$ and $\Phi$ in previous work are optimized in the same set of training data, $\Phi$ would choose the module with large complexity to overfit these data. In contrast, SN essentially prevents overfitting by choosing normalizers to improve both learning and generalization ability as discussed below.

**Analyses of SN.** To understand SN, we theoretically compare SN with BN, IN, and LN by representing them using weight normalization (WN) (Salimans & Kingma, 2016) that is independent of mean and variance. WN is computed as $v\frac{\mathbf{w}^\mathsf{T}\mathbf{x}}{\|\mathbf{w}\|_2}$, where $\mathbf{w}$ and $\mathbf{x}$ represent a filter and an image patch. WN normalizes the norm of each filter to 1 and rescales to $v$.

**Remark 1.** *Let $\mathbf{x}$ be an image patch with zero mean and unit variance, $\mathbf{w}_i$ be a filter of the $i$-th channel, $i \in \{1, 2, ..., C\}$, and $v$ be the filter norm of WN. Eqn.(1) can be rewritten by $\hat{h}_{\mathrm{in}} = \gamma\frac{\mathbf{w}_i^\mathsf{T}\mathbf{x}}{\|\mathbf{w}_i\|_2} + \beta$; $\hat{h}_{\mathrm{bn}} = \gamma\frac{\mathbf{w}_i^\mathsf{T}\mathbf{x}}{\|\mathbf{w}_i\|_2} + \beta$, s.t. $\gamma \leq v$; and $\hat{h}_{\mathrm{ln}} = \gamma\frac{\mathbf{w}_i^\mathsf{T}\mathbf{x}}{\|\mathbf{w}_i\|_2 + \sum_{j\neq i}^C \|\mathbf{w}_j\|_2} + \beta$. By combining them, SN in Eqn.(3) can be reformulated by $\hat{h}_{\mathrm{sn}} = w_{\mathrm{in}}\hat{h}_{\mathrm{in}} + w_{\mathrm{bn}}\hat{h}_{\mathrm{bn}} + w_{\mathrm{ln}}\hat{h}_{\mathrm{ln}} = \gamma\frac{\mathbf{w}_i^\mathsf{T}\mathbf{x}}{\|\mathbf{w}_i\|_2 + w_{\mathrm{ln}}\sum_{j\neq i}^C \|\mathbf{w}_j\|_2} + \beta$, s.t. $w_{\mathrm{bn}}\gamma \leq v$, where $w_{\mathrm{in}}, w_{\mathrm{ln}}, w_{\mathrm{bn}}$ are the weights and $\gamma \leq v$ is a constraint.*

Remark 1 simplifies SN in Eqn.(3), enabling us to compare different normalizers geometrically by formulating them with respect to WN. In Fig.3, $\hat{h}_{\mathrm{in}}$ of IN can be computed similarly to WN with an additional bias $\beta$, where the norms of all filters are normalized to 1 and then rescaled to $\gamma$. As $\gamma$ and $v$ have the same learning dynamic, the length of $\gamma$ would be identically to $v$ (see $\mathbf{w}_1, \mathbf{w}_2$ of IN). Moreover, $\hat{h}_{\mathrm{bn}}$ in BN can be rewritten as WN with regularization over $\gamma$, making it shorter than $v$. Compared to IN and LN, Luo et al. (2019) shows that the regularization of BN improves generalization and increases angle between filters, preventing them from coadaptation (see $\mathbf{w}_1, \mathbf{w}_2$ of BN). Furthermore, $\hat{h}_{\mathrm{ln}}$ in LN normalizes each filter among channels where the filter norm is less constrained than IN and BN. That is, LN allows $\gamma > v$ to increase learning ability. Finally, $\hat{h}_{\mathrm{sn}}$ in SN inherits the benefits from all of them and enables balance between

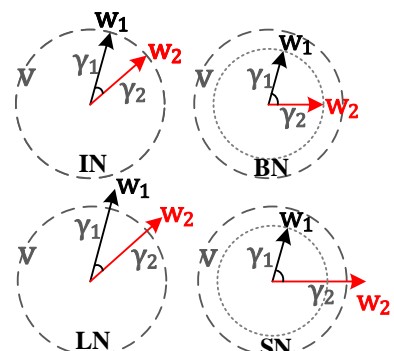

Figure 3: Geometric view of directions and lengths of the filters in IN, BN, LN, and SN by comparing them to WN.

learning and generalization ability. For example, when the batch size is small, the random noise from the batch statistics of BN would be too strong. SN is able to maintain performance by decreasing $w_{\mathrm{bn}}$ and increasing $w_{\mathrm{ln}}$, such that the regularization from BN is reduced and the learning ability is enhanced by LN. This phenomenon is supported by our experiment. More results are provided in Appendix B.

**Variants of SN.** SN has many extensions. For instance, a pretrained network with SN can be fine-tuned by applying the `argmax` function on its control parameters where each normalization layer selects only one normalizer, leading to sparse SN. For $(8, 32)$ as an example, SN with sparsity achieves top-1 accuracy of 77.0% in ImageNet with ResNet50, which is comparable to 76.9% of SN without sparsity. Moreover, when the channels are divided into groups, each group could select its own normalizer to increase representation power of SN. Our preliminary results suggest that group SN performs better than SN in some senses. For instance, group SN with only two groups boosts the top-1 accuracy of ResNet50 to 77.2% in ImageNet. The above two variants will be presented as future work due to the length of paper. This work focuses on SN where the importance weights are tied between channels.

**Inference.** When applying SN in test, the statistics of IN and LN are computed independently for each sample, while BN uses batch average *after* training without computing moving average in each iteration. Here batch average is performed in two steps. First, we freeze the parameters

| | Parameter | | | Statistical Estimation | | |
|---|---|---|---|---|---|---|
| | params | #params | hyper-params | statistics | computation complexity | #statistics |
| BN (Ioffe & Szegedy, 2015) | $\gamma, \beta$ | $2C$ | $p, \epsilon$ | $\mu, \sigma, \mu', \sigma'$ | $\mathcal{O}(NCHW)$ | $2C$ |
| IN (Ulyanov et al., 2016) | $\gamma, \beta$ | $2C$ | $\epsilon$ | $\mu, \sigma$ | $\mathcal{O}(NCHW)$ | $2CN$ |
| LN (Ba et al., 2016) | $\gamma, \beta$ | $2C$ | $\epsilon$ | $\mu, \sigma$ | $\mathcal{O}(NCHW)$ | $2N$ |
| GN (Wu & He, 2018) | $\gamma, \beta$ | $2C$ | $g, \epsilon$ | $\mu, \sigma$ | $\mathcal{O}(NCHW)$ | $2gN$ |
| BRN (Ioffe, 2017) | $\gamma, \beta$ | $2C$ | $p, \epsilon, r, d$ | $\mu, \sigma, \mu', \sigma'$ | $\mathcal{O}(NCHW)$ | $2C$ |
| BKN (Wang et al., 2018) | $A$ | $C^2$ | $p, \epsilon$ | $\mu, \Sigma, \mu', \Sigma'$ | $\mathcal{O}(NC^2HW)$ | $C + C^2$ |
| WN (Salimans & Kingma, 2016) | $\gamma$ | $C$ | $-$ | $-$ | $-$ | $-$ |
| SN | $\gamma, \beta,$ $\{w_k\}_{k \in \Omega}$ | $2C + 6$ | $\epsilon$ | $\{\mu_k, \sigma_k\}_{k \in \Omega}$ | $\mathcal{O}(NCHW)$ | $2C + 2N$ $+2CN$ |

Table 1: **Comparisons of normalization methods**. First, we compare their types of parameters, numbers of parameters (#params), and hyper-parameters. Second, we compare types of statistics, computational complexity to estimate statistics, and numbers of statistics (#statistics). Specifically, $\gamma, \beta$ denote the scale and shift parameters. $\mu, \sigma, \Sigma$ are a vector of means, a vector of standard deviations, and a covariance matrix. $\mu'$ represents the moving average. Moreover, $p$ is the momentum of moving average, $g$ in GN is the number of groups, $\epsilon$ is a small value for numerical stability, and $r, d$ are used in BRN. In SN, $k \in \Omega$ indicates a set of different kinds of statistics, $\Omega = \{\text{in}, \text{ln}, \text{bn}\}$, and $w_k$ is an importance weight of each kind.

of the network and all the SN layers, and feed the network with a certain number of mini-batches randomly chosen from the training set. Second, we average the means and variances produced by all these mini-batches in each SN layer. The averaged statistics are used by BN in SN.

We find that batch average makes training converged faster than moving average. It can be computed by using a small amount of samples. For example, top-1 accuracies of ResNet50 on ImageNet by using batch average with 50k and all training samples are 76.90% and 76.92% respectively. They are trained much faster and slightly better than 76.89% of moving average. Appendix A shows more results.

**Implementation.** SN can be easily implemented in existing softwares such as PyTorch and Tensor-Flow. The backward computation of SN can be obtained by automatic differentiation (AD) in these softwares. Without AD, we need to implement back-propagation (BP) of SN, where the errors are propagated through $\mu_k$ and $\sigma_k^2$. We provide the derivations of BP in Appendix H.

## 3  RELATIONSHIPS TO PREVIOUS WORK

In Table 1, we compare SN to BN, IN, LN, and GN, as well as three variants of BN including Batch Renormalization (BRN), Batch Kalman Normalization (BKN), and WN. In general, we see that SN possesses comparable numbers of parameters and computations, as well as rich statistics. Details are presented below.

• First, SN has similar number of parameters compared to previous methods, as shown in the first portion of Table 1. Most of the approaches learn a scale parameter $\gamma$ and a bias $\beta$ for each one of the $C$ channels, resulting in $2C$ parameters. SN learns 6 importance weights as the additional parameters. We see that BKN has the maximum number of $C^2$ parameters, as it learns a transformation matrix $A$ for the means and variances. WN has $C$ scale parameters without the biases.

Furthermore, many methods have $p$ and $\epsilon$ as hyper-parameters, whose values are not sensitive because they are often fixed in different tasks. In contrast, GN and BRN have to search the number of groups $g$ or the renormalization parameters $r, d$, which may have different values in different networks. Moreover, WN does not have hyper-parameters and statistics, since it performs normalization in the space of network parameters rather than feature space. Salimans & Kingma (2016); Luo et al. (2019) showed that WN is a special case of BN.

• Second, although SN has richer statistics, the computational complexity to estimate them is comparable to previous methods, as shown in the second portion of Table 1. As introduced in Sec.2, IN, LN, and BN estimate the means and variances along axes $(H, W)$, $(C, H, W)$, and $(N, H, W)$ respectively, leading to $2CN$, $2N$, and $2C$ numbers of statistics. Therefore, SN has $2CN + 2N + 2C$ statistics by combining them. Although BKN has the largest number of $C + C^2$ statistics, it also has the highest computations because it estimates the covariance matrix other than the variance vector. Also, approximating the covariance matrix in a minibatch is nontrivial as discussed in (Desjardins et al., 2015; Luo, 2017b;a). BN, BRN, and BKN also compute moving averages.

• Third, SN is demonstrated in various networks, tasks, and datasets. Its applications are much wider than existing normalizers and it also has rich theoretical value that is worth exploring.

We would also like to acknowledge the contributions of previous work that explored spatial region (Ren et al., 2016) and conditional normalization (Perez et al., 2017).

# 4 EXPERIMENTS

This section presents the main results of SN in multiple challenging problems and benchmarks, such as ImageNet (Russakovsky et al., 2015), COCO (Lin et al., 2014), Cityscapes (Cordts et al., 2016), ADE20K (Zhou et al., 2017), and Kinetics (Kay et al., 2017), where the effectiveness of SN is demonstrated by comparing with existing normalization techniques.

## 4.1 IMAGE CLASSIFICATION IN IMAGENET

SN is first compared with existing normalizers on the ImageNet classification dataset of 1k categories. All the methods adopt ResNet50 as backbone network. The experimental setting and more results are given in Appendix C.

**Comparisons.** The top-1 accuracy on the 224×224 center crop is reported for all models. SN is compared to BN and GN as shown in Table 2. In the first five columns, we see that the accuracy of BN reduces by 1.1% from $(8, 16)$ to $(8, 8)$ and declines to 65.3% of $(8, 2)$, implying that BN is unsuitable in small minibatch, where the random noise from the statistics is too heavy. GN obtains around 75.9% in all cases, while SN outperforms BN and GN in almost all cases, rendering its robustness to different batch sizes. In Appendix, Fig.6 plots the training and validation curves, where SN enables faster convergence while maintains higher or comparable accuracies than those of BN and GN.

The middle two columns of Table 2 average the gradients in a single GPU by using only 16 and 32 samples, such that their batch sizes are the same as $(8, 2)$ and $(8, 4)$. SN again performs best in these single-GPU settings, while BN outperforms GN. For example, unlike $(8, 4)$ that uses 8 GPUs, BN achieves 76.5% in $(1, 32)$, which is the best-performing result of BN, although the batch size to compute the gradients is as small as 32. From the above results, we see that BN's performance are sensitive to the statistics more than the gradients, while SN are robust to both of them. The last two columns of Table 2 have the same batch size of 8, where $(1, 8)$ has a minibatch size of 8, while $(8, 1)$ is an extreme case with a single sample in a minibatch. For $(1, 8)$, SN performs best. For $(8, 1)$, SN consists of IN and LN but no BN, because IN and BN are the same in training when the minibatch size is 1. In this case, both SN and GN still perform reasonably well, while BN fails to converge.

**Ablation Study.** Fig.1 (a) and Fig.4 plot histograms to compare the importance weights of SN with respect to different tasks and batch sizes. These histograms are computed by averaging the importance weights of all SN layers in a network. They show that SN adapts to various scenarios by changing its importance weights. For example, SN prefers BN when the minibatch is sufficiently large, while it selects LN instead when small minibatch is presented, as shown in the green and red bars of Fig.4. These results are in line with our analyses in Sec.2.1.

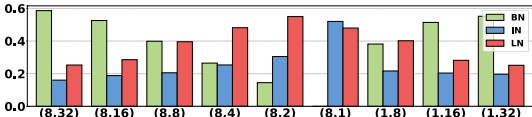

Figure 4: Importance weights *v.s.* batch sizes. The bracket $(\cdot, \cdot)$ indicates (#GPUs, #samples per GPU). SN doesn't have BN in $(8, 1)$.

Furthermore, we repeat training of ResNet50 several times in ImageNet, to show that when the network, task, batch setting and data are fixed, the importance weights of SN are not sensitive to the change of training protocols such as solver, parameter initialization, and learning rate decay. As a result, we find that all trained models share similar importance weights.

The importance weights in each SN layer are visualized in Appendix C.2. Overall, examining the selectivity of SN layers discloses interesting characteristics and impacts of normalization methods in deep learning, and sheds light on model design in many research fields.

|        | (8,32) | (8,16) | (8,8) | (8,4) | (8,2) | (1,16) | (1,32) | (8,1) | (1,8) |
|--------|--------|--------|-------|-------|-------|--------|--------|-------|-------|
| BN     | 76.4   | 76.3   | 75.2  | 72.7  | 65.3  | 76.2   | 76.5   | –     | 75.4  |
| GN     | 75.9   | 75.8   | 76.0  | 75.8  | **75.9** | 75.9 | 75.8   | **75.5** | 75.5 |
| SN     | **76.9** | **76.7** | **76.7** | 75.9 | 75.6 | **76.3** | **76.6** | 75.0 | **75.9** |
| GN−BN  | -0.5   | -0.5   | 0.8   | 3.1   | 10.6  | -0.3   | -0.7   | –     | 0.1   |
| SN−BN  | 0.5    | 0.4    | 1.5   | 3.2   | 10.3  | 0.1    | 0.1    | –     | 0.5   |
| SN−GN  | 1.0    | 0.9    | 0.7   | 0.1   | -0.3  | 0.4    | 0.8    | -0.5  | 0.4   |

Table 2: **Comparisons of top-1 accuracies** on the validation set of ImageNet, by using ResNet50 trained with SN, BN, and GN in different batch size settings. The bracket $(\cdot, \cdot)$ denotes (#GPUs, #samples per GPU). In the bottom part, 'GN-BN' indicates the difference between the accuracies of GN and BN. The '-' in $(8, 1)$ indicates BN does not converge. The best-performing result of each setting is shown in bold.

| backbone | head | AP | $AP_{.5}$ | $AP_{.75}$ | $AP_l$ | $AP_m$ | $AP_s$ |
|----------|------|------|------|------|------|------|------|
| BN[†]    | –    | 36.7 | 58.4 | 39.6 | 48.1 | 39.8 | 21.1 |
| BN[†]    | GN   | 37.2 | 58.0 | 40.4 | 48.6 | 40.3 | 21.6 |
| BN[†]    | SN   | 38.0 | 59.4 | 41.5 | 48.9 | 41.3 | 22.7 |
| GN       | GN   | 38.2 | 58.7 | 41.3 | 49.6 | 41.0 | 22.4 |
| SN       | SN   | **39.3** | **60.9** | **42.8** | **50.3** | **42.7** | **23.5** |

| backbone | head | $AP^b$ | $AP^b_{.5}$ | $AP^b_{.75}$ | $AP^m$ | $AP^m_{.5}$ | $AP^m_{.75}$ |
|----------|------|------|------|------|------|------|------|
| BN[†]    | –    | 38.6 | 59.5 | 41.9 | 34.2 | 56.2 | 36.1 |
| BN[†]    | GN   | 39.5 | 60.0 | 43.2 | 34.4 | 56.4 | 36.3 |
| BN[†]    | SN   | 40.0 | 61.0 | 43.3 | 34.8 | 57.3 | 36.3 |
| GN       | GN   | 40.2 | 60.9 | 43.8 | 35.7 | 57.8 | 38.0 |
| GN       | SN   | 40.4 | 61.4 | 44.2 | 36.0 | 58.4 | 38.1 |
| SN       | SN   | **41.0** | **62.3** | **45.1** | **36.5** | **58.9** | **38.7** |

Table 3: **Faster R-CNN+FPN** using ResNet50 and FPN with 1x LR schedule. BN[†] represents BN is frozen. The best results are bold.

Table 4: **Mask R-CNN** using ResNet50 and FPN with 2x LR schedule. BN[†] represents BN is frozen without finetuning. The best results are bold.

**SN v.s. IN and LN.** IN and LN are not optimal in image classification as reported in (Ulyanov et al., 2016) and (Ba et al., 2016). With a regular setting of $(8, 32)$, ResNet50 trained with IN and LN achieve 71.6% and 74.7% respectively, which reduce 5.3% and 2.2% compared to 76.9% of SN.

**SN v.s. BRN and BKN.** BRN has two extra hyper-parameters, $r_{max}$ and $d_{max}$, which renormalize the means and variances. We choose their values as $r_{max} = 1.5$ and $d_{max} = 0.5$, which work best for ResNet50 in the setting of $(8, 4)$ following (Ioffe, 2017). 73.7% of BRN surpasses 72.7% of BN by 1%, but it reduces 2.2% compared to 75.9% of SN.

BKN (Wang et al., 2018) estimated the statistics in the current layer by combining those computed in the preceding layers. It estimates the covariance matrix rather than the variance vector. In particular, how to connect the layers requires careful design for every specific network. For ResNet50 with $(8, 32)$, BKN achieved 76.8%, which is comparable to 76.9% of SN. However, for small minibatch, BKN reported 76.1% that was evaluated in a micro-batch setting where 256 samples are used to compute gradients and 4 samples to estimate the statistics. This setting is easier than $(8, 4)$ that uses 32 samples to compute gradients. Furthermore, it is unclear how to apply BRN and BKN in the other tasks such as object detection and segmentation.

## 4.2 OBJECT DETECTION AND INSTANCE SEGMENTATION IN COCO

Next we evaluate SN in object detection and instance segmentation in COCO (Lin et al., 2014). Unlike image classification, these two tasks benefit from large size of input images, making large memory footprint and therefore leading to small minibatch size, such as 2 samples per GPU (Ren et al., 2015; Lin et al., 2016). In this case, as BN is not applicable in small minibatch, previous work (Ren et al., 2015; Lin et al., 2016; He et al., 2017) often freeze BN and turns it into a constant linear transformation layer, which actually performs no normalization. Overall, SN selects different operations in different components of a detection system (see Fig.1), showing much more superiority than both BN and GN. The experimental settings and more results are given in Appendix D.

Table 3 reports results of Faster R-CNN by using ResNet50 and the Feature Pyramid Network (FPN) (Lin et al., 2016). A baseline BN[†] achieves an AP of 36.7 without using normalization in the detection head. When using SN and GN in the head and BN[†] in the backbone, BN[†]+SN improves the AP of BN[†]+GN by 0.8 (from 37.2 to 38.0). We investigate using SN and GN in both the backbone and head. In this case, we find that GN improves BN[†]+SN by only a small margin of 0.2 AP (38.2 v.s. 38.0), although the backbone is pretrained and finetuned by using GN. When finetuning the SN backbone, SN obtains a significant improvement of 1.1 AP over GN (39.3 v.s. 38.2). Furthermore, the 39.3 AP of SN and 38.2 of GN both outperform 37.8 in (Peng et al., 2017), which synchronizes BN layers in the backbone (i.e. BN layers are not frozen).

|  | ADE20K | | Cityscapes | |
|---|---|---|---|---|
|  | $\text{mIoU}_{\text{ss}}$ | $\text{mIoU}_{\text{ms}}$ | $\text{mIoU}_{\text{ss}}$ | $\text{mIoU}_{\text{ms}}$ |
| SyncBN | 36.4 | 37.7 | 69.7 | 73.0 |
| GN | 35.7 | 36.3 | 68.4 | 73.1 |
| SN | **38.7** | **39.2** | **71.2** | **75.1** |

|  | batch=8, length=32 | | batch=4, length=32 | |
|---|---|---|---|---|
|  | top1 | top5 | top1 | top5 |
| BN | 73.2 | 90.9 | 72.1 | 90.0 |
| GN | 73.0 | 90.6 | 72.8 | 90.6 |
| SN | **73.5** | **91.3** | **73.3** | **91.2** |

Table 5: **Results in ADE20K validation set and Cityscapes test set** by using ResNet50 with dilated convolutions. 'ss' and 'ms' indicate single-scale and multi-scale inference. SyncBN represents mutli-GPU synchronization of BN. SN finetunes from $(8, 2)$ pretrained model.

Table 6: **Results of Kinetics dataset.** In training, the clip length of 32 frames is regularly sampled with a frame interval of 2. We study a batch size of 8 or 4 clips per GPU. BN is not synchronized across GPUs. SN finetunes from $(8, 2)$ pretrained model.

Table 4 reports results of Mask R-CNN (He et al., 2017) with FPN. In the upper part, SN is compared to a head with no normalization and a head with GN, while the backbone is pretrained with BN, which is then frozen in finetuning (*i.e.* the ImageNet pretrained features are the same). We see that the baseline BN[†] achieves a box AP of 38.6 and a mask AP of 34.2. SN improves GN by 0.5 box AP and 0.4 mask AP, when finetuning the same BN[†] backbone.

More direct comparisons with GN are shown in the lower part of Table 4. We apply SN in the head and finetune the same backbone network pretrained with GN. In this case, SN outperforms GN by 0.2 and 0.3 box and mask APs respectively. Moreover, when finetuning the SN backbone, SN surpasses GN by a large margin of both box and mask AP (41.0 *v.s.* 40.2 and 36.5 *v.s.* 35.7). Note that the performance of SN even outperforms 40.9 and 36.4 of the 101-layered ResNet (Girshick et al., 2018).

### 4.3 Semantic Image Parsing in Cityscapes and ADE20K

We investigate SN in semantic image segmentation in ADE20K (Zhou et al., 2017) and Cityscapes (Cordts et al., 2016). The empirical setting can be found in Appendix E.

Table 5 reports mIoU on the ADE20K validation set and Cityscapes test set, by using both single-scale and multi-scale testing. In SN, BN is not synchronized across GPUs. In ADE20K, SN outperforms SyncBN with a large margin in both testing schemes (38.7 *v.s.* 36.4 and 39.2 *v.s.* 37.7), and improve GN by 3.0 and 2.9. In Cityscapes, SN also performs best compared to SyncBN and GN. For example, SN surpasses SyncBN by 1.5 and 2.1 in both testing scales. We see that GN performs worse than SyncBN in these two benchmarks. Fig.9 in Appendix compares the importance weights of SN in ResNet50 trained on both ADE20K and Cityscapes, showing that different datasets would choose different normalizers when the models and tasks are the same.

### 4.4 Video Recognition in Kinetics

We evaluate video recognition in Kinetics dataset (Kay et al., 2017), which has 400 action categories. We experiment with Inflated 3D (I3D) convolutional networks (Carreira & Zisserman, 2017) and employ the ResNet50 I3D baseline as described in (Wu & He, 2018). The models are pretrained from ImageNet. For all normalizers, we extend the normalization from over $(H, W)$ to over $(T, H, W)$, where $T$ is the temporal axis. We train in the training set and evaluate in the validation set. The top1 and top5 classification accuracy are reported by using standard 10-clip testing that averages softmax scores from 10 clips sampled regularly.

Table 6 shows that SN works better than BN and GN in both batch sizes. For example, when batch size is 4, top1 accuracy of SN is better than BN and GN by 1.2% and 0.5%. It is seen that SN already surpasses BN and GN with batch size of 8. SN with batch size 8 further improves the results.

### 4.5 On the other Tasks

We also evaluate SN in the tasks of artistic image stylization (Johnson et al., 2016) and efficient neural architecture search (Pham et al., 2018). The results are presented in Appendix F and G, where SN achieves competitive results.

## 5 DISCUSSIONS AND FUTURE WORK

This work presented Switchable Normalization (SN) to learn different operations in different normalization layers of a deep network. This novel perspective opens up new direction in many research fields that employ deep learning, such as CV, ML, NLP, Robotics, and Medical Imaging. This work has demonstrated SN in multiple tasks of CV such as recognition, detection, segmentation, image stylization, and neural architecture search, where SN outperforms previous normalizers without bells and whistles. The implementations of these experiments will be released. Our analyses (Luo et al., 2018) suggest that SN has an appealing characteristic to balance learning and generalization when training deep networks. Investigating SN facilitates the understanding of normalization approaches (Shao et al., 2019; Pan et al., 2019; Luo, 2017a;b), such as sparse SN (Shao et al., 2019) and switchable whitening (Pan et al., 2019).

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

## APPENDICES

## A INFERENCE OF SN

In SN, BN employs batch average rather than moving average. We provide comparisons between them as shown in Fig.5, where SN is evaluated with both moving average and batch average to estimate the statistics used in test. They are used to train ResNet50 on ImageNet. The two settings of SN produce similar results of 76.9% when converged, which is better than 76.4% of BN. We see that SN with batch average converges faster and more stably than BN and SN that use moving average. In this work, we find that for all batch settings, SN with batch average provides results better than moving average. We also found that the conventional BN can be improved by replacing moving average by using batch average.

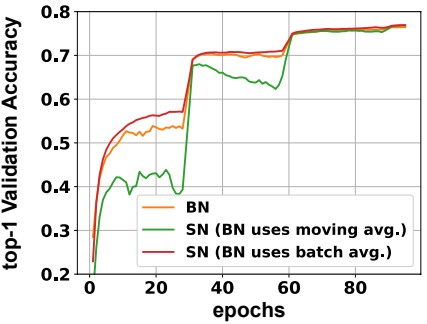

Figure 5: Comparisons of 'BN', 'SN with moving average', and 'SN with batch average', when training ResNet50 on ImageNet in $(8, 32)$. We see that SN with batch average produces faster and more stable convergence than the other methods.

## B PROOF OF REMARK 1

**Remark 1.** *Suppose $\mathbf{x}$ to be an image patch with zero mean and unit variance, $\mathbf{w}_i$ to be a filter of the $i$-th channel, $i \in \{1, 2, ..., C\}$, and $v$ to be the filter norm of WN. Eqn.(1) can be rewritten as $\hat{h}_{\mathrm{in}} = \gamma \frac{\mathbf{w}_i^\mathsf{T}\mathbf{x}}{\|\mathbf{w}_i\|_2} + \beta$; $\hat{h}_{\mathrm{bn}} = \gamma \frac{\mathbf{w}_i^\mathsf{T}\mathbf{x}}{\|\mathbf{w}_i\|_2} + \beta$, s.t. $\gamma \leq v$; and $\hat{h}_{\mathrm{ln}} = \gamma \frac{\mathbf{w}_i^\mathsf{T}\mathbf{x}}{\|\mathbf{w}_i\|_2 + \sum_{j \neq i}^C \|\mathbf{w}_j\|_2} + \beta$. By combining them, Eqn.(3) is rewritten as $\hat{h}_{\mathrm{sn}} = \gamma \frac{\mathbf{w}_i^\mathsf{T}\mathbf{x}}{\|\mathbf{w}_i\|_2 + w_{\mathrm{ln}}\sum_{j \neq i}^C \|\mathbf{w}_j\|_2} + \beta$, s.t. $w_{\mathrm{bn}}\gamma \leq v$, where $w_{\mathrm{ln}}, w_{\mathrm{bn}}$ are the weights and $\gamma \leq v$ is a constraint.*

*Proof.* Eqn.(1) shows that IN, LN, and BN can be generally computed as $\hat{h}_k = \gamma \frac{\mathbf{w}_i^\mathsf{T}\mathbf{x} - \mu_k}{\sigma_k} + \beta$, $k \in \{\mathrm{in}, \mathrm{ln}, \mathrm{bn}\}$. When $\mathbf{x}$ is normalized to zero mean and unit variance, we have $\hat{h}_{\mathrm{in}} = \gamma \frac{\mathbf{w}_i^\mathsf{T}\mathbf{x}}{\|\mathbf{w}_i\|_2} + \beta$ and $\hat{h}_{\mathrm{ln}} = \gamma \frac{\mathbf{w}_i^\mathsf{T}\mathbf{x}}{\|\mathbf{w}_i\|_2 + \sum_{j \neq i}^C \|\mathbf{w}_j\|_2} + \beta$ according to their definitions.

For BN, we follow the derivations in (Luo et al., 2019) where the batch statistics $\mu_{\mathrm{bn}}$ and $\sigma_{\mathrm{bn}}$ are treated as random variables. BN can be reformulated as population normalization (PN) and adaptive gamma decay. Let $\mathcal{L} = \frac{1}{P}\sum_{j=1}^P \mathbb{E}_{\mu_{\mathrm{bn}}, \sigma_{\mathrm{bn}}} \ell(\hat{h}_{\mathrm{bn}}^j)$ be the expected loss function of BN by integrating over random variables $\mu_{\mathrm{bn}}$ and $\sigma_{\mathrm{bn}}$. We have $\mathcal{L} \simeq \frac{1}{P}\sum_{j=1}^P \ell(\bar{h}_{\mathrm{pn}}^j) + \zeta(h)\gamma^2$, where $\bar{h}_{\mathrm{pn}}^j = \gamma \frac{h^j - \mu_\mathcal{P}}{\sigma_\mathcal{P}} + \beta$ represents population normalization (PN) with $h^j = \mathbf{w}^\mathsf{T}\mathbf{x}^j$. $\mu_\mathcal{P}$ and $\sigma_\mathcal{P}$ are the population mean and population standard deviation. $\zeta(h)$ is a data-dependent coefficient. Therefore, $\zeta(h)\gamma^2$ represents adaptive gamma regularization whose strength is depended on training data. With normalized input, we have $\mu_\mathcal{P} = 0$ and $\sigma_\mathcal{P} = 1$. Thus PN can be rewritten as WN, that is, $\bar{h}_{\mathrm{pn}} = \gamma \frac{\mathbf{w}_i^\mathsf{T}\mathbf{x}}{\|\mathbf{w}_i\|_2} + \beta$. Let WN be defined as $v\frac{\mathbf{w}_i^\mathsf{T}\mathbf{x}}{\|\mathbf{w}_i\|_2}$. Then $v$ in WN and $\gamma$ in PN have the same learning dynamic. However, the adaptive gamma regularization imposes the $\gamma \leq v$ constraint to BN, since WN does not have regularization on $v$. Compared to WN, we express BN as $\hat{h}_{\mathrm{bn}} = \gamma \frac{\mathbf{w}_i^\mathsf{T}\mathbf{x}}{\|\mathbf{w}_i\|_2} + \beta$, s.t. $\gamma \leq v$. □

## C IMAGENET

### C.1 EXPERIMENTAL SETTING

All models in ImageNet are trained on 1.2M images and evaluated on 50K validation images. They are trained by using SGD with different settings of batch sizes, which are denoted as a 2-tuple, (*number of GPUs, number of samples per GPU*). For each setting, the gradients are aggregated over all GPUs, and the means and variances of the normalization methods are computed in each

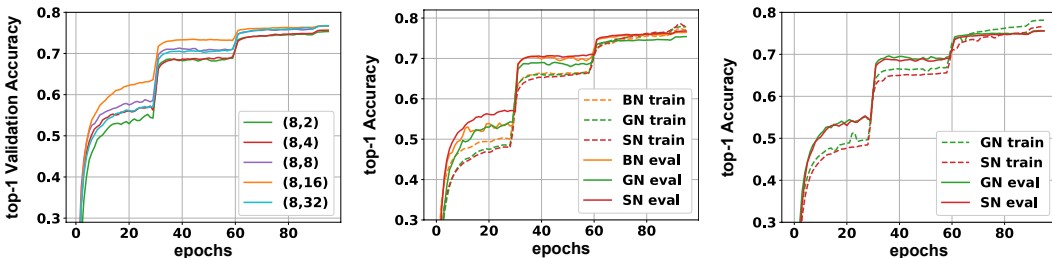

(a) validation curves of SN in different batch sizes.

(b) train and validation curves of (8,32), SN *v.s.* BN and GN.

(c) train and validation curves of (8,2), SN *v.s.* GN.

Figure 6: **Comparisons of learning curves.** (a) visualizes the validation curves of SN with different settings of batch size. The bracket $(\cdot,\cdot)$ denotes (#GPUs, #samples per GPU). (b) compares the top-1 train and validation curves on ImageNet of SN, BN, and GN in the batch size of (8,32). (c) compares the train and validation curves of SN and GN in the batch size of (8,2).

GPU. The network parameters are initialized by following (He et al., 2016). For all normalization methods, all $\gamma$'s are initialized as 1 and all $\beta$'s as 0. The parameters of SN ($\lambda_k$ and $\lambda'_k$) are initialized as 1. We use a weight decay of $10^{-4}$ for all parameters including $\gamma$ and $\beta$. All models are trained for 100 epoches with a initial learning rate of 0.1, which is deceased by $10\times$ after 30, 60, and 90 epoches. For different batch sizes, the initial learning rate is linearly scaled according to (Goyal et al., 2017). During training, we employ data augmentation the same as (He et al., 2016). The top-1 classification accuracy on the $224\times224$ center crop is reported.

## C.2 MORE RESULTS

Fig.6 (a) plots the validation curves of SN. Fig.6 (b) and (c) compare the training and validation curves of SN, BN and GN in $(8,32)$ and $(8,2)$ respectively. From all these curves, we see that SN enables faster convergence while maintains higher or comparable accuracies than those of BN and GN.

**Ablation Study of Importance Weights.** In particular, the selected operations of each SN layer are shown in Fig.7. We have several observations. First, for the same batch size, the importance weights of $\mu$ and $\sigma$ could have notable differences, especially when comparing 'res1,4,5' of (a,b) and 'res2,4,5' of (c,d). For example, $\sigma$ of BN (green) in 'res5' in (b,d) are mostly reduced compared to $\mu$ of BN in (a,c). As discussed in (Ioffe & Szegedy, 2015; Salimans & Kingma, 2016), this is because the variance estimated in a minibatch produces larger noise than the mean, making training instable. SN is able to restrain the noisy statistics and stabilize training.

Second, the SN layers in different places of a network may select distinct operations. In other words, when comparing the adjacent SN layers after the $3 \times 3$ conv layer, shortcut, and the $1 \times 1$ conv layer, we see that they may choose different importance weights, *e.g.* 'res2,3'. The selectivity of operations in different places (normalization layers) of a deep network has not been observed in previous work.

Third, deeper layers prefer LN and IN more than BN, as illustrated in 'res5', which tells us that putting BN in an appropriate place is crucial in the design of network architecture. Although the stochastic uncertainty in BN (*i.e.* the minibatch statistics) acts as a regularizer that might benefit generalization, using BN uniformly in all normalization layers may impede performance.

## D COCO DATASET

SN is easily plugged into different detection frameworks implemented by using different softwares. We implement it on existing detection softwares of PyTorch and Caffe2-Detectron (Girshick et al., 2018) respectively. We conduct 3 settings, including **setting-1:** Faster R-CNN (Ren et al., 2015) on PyTorch; **setting-2:** Faster R-CNN+FPN (Lin et al., 2016) on Caffe2; and **setting-3:** Mask R-CNN (He et al., 2017)+FPN on Caffe2. For all these settings, we choose ResNet50 as the backbone network. In each setting, the experimental configurations of all the models are the same, while only the normalization layers are replaced. All models of SN are finetuned from $(8, 2)$ in ImageNet.

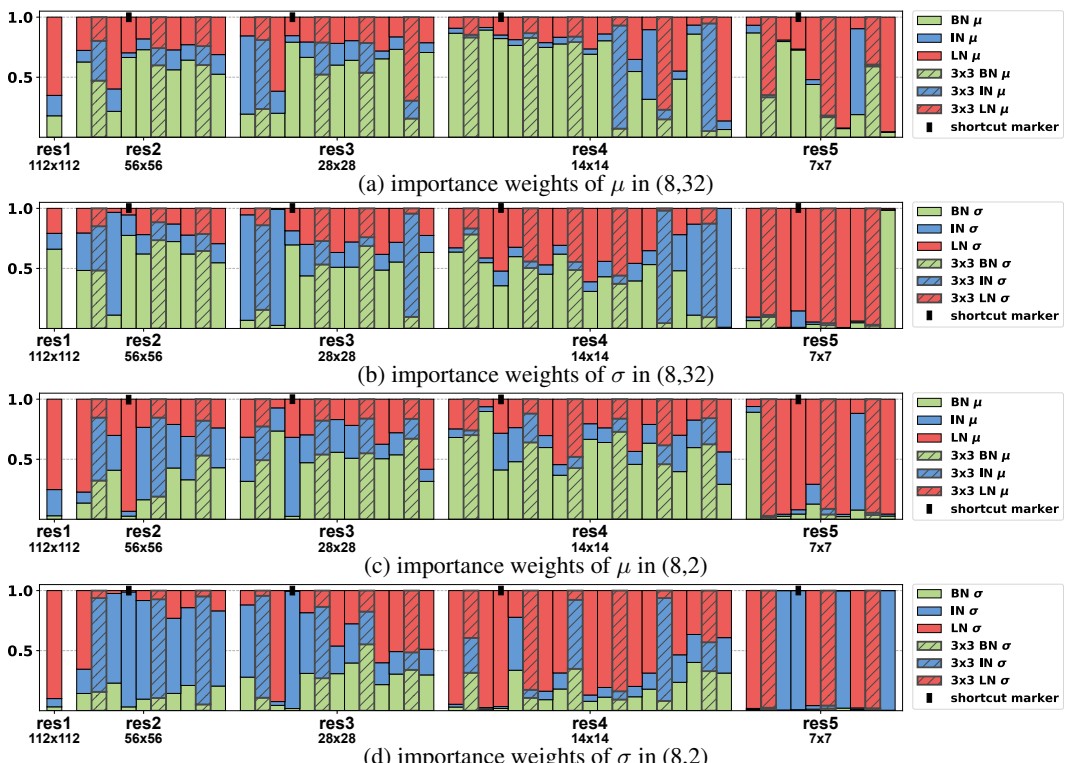

Figure 7: **Selected operations of each SN layer in ResNet50.** There are 53 SN layers. (a,b) show the importance weights for $\mu$ and $\sigma$ of $(8, 32)$, while (c,d) show those of $(8, 2)$. The $y$-axis represents the importance weights that sum to 1, while the $x$-axis shows different residual blocks of ResNet50. The SN layers in different places are highlighted differently. For example, the SN layers follow the $3 \times 3$ conv layers are outlined by shaded color, those in the shortcuts are marked with '■', while those follow the $1 \times 1$ conv layers are in flat color. The first SN layer follows a $7 \times 7$ conv layer. We see that SN learns distinct importance weights for different normalization methods as well as $\mu$ and $\sigma$, adapting to different batch sizes, places, and depths of a deep network.

| backbone | head | AP | AP$_{.5}$ | AP$_{.75}$ | AP$_l$ | AP$_m$ | AP$_s$ |
|----------|------|------|------|------|------|------|------|
| BN$^\dagger$ | BN$^\dagger$ | 29.6 | 47.8 | 31.9 | 45.5 | 33.0 | 11.5 |
| BN | BN | 19.3 | 33.0 | 20.0 | 32.3 | 21.3 | 7.4 |
| GN | GN | 32.7 | 52.4 | 35.1 | **49.1** | 36.1 | 14.9 |
| SN | SN | **33.0** | **52.9** | **35.7** | 48.7 | **37.2** | **15.6** |
| BN$^\ddagger$ | BN | 20.0 | 33.5 | 21.1 | 32.1 | 21.9 | 7.3 |
| GN$^\ddagger$ | GN | 28.3 | 46.3 | 30.1 | 41.2 | 30.0 | 12.7 |
| SN$^\ddagger$ | SN | **29.5** | **47.8** | **31.6** | **44.2** | **32.6** | **13.0** |

Table 7: **Faster R-CNN for detection in COCO** using ResNet50 and RPN. BN$^\dagger$ represents BN is frozen without finetuning. The superscript '$\ddagger$' indicates the backbones are trained from scratch without pretraining on ImageNet.

**Experimental Settings.** For **setting-1**, we employ a fast implementation (Yang et al., 2017) of Faster R-CNN in PyTorch and follow its protocol. Specifically, we train all models on 4 GPUs and 3 images per GPU. Each image is re-scaled such that its shorter side is 600 pixels. All models are trained for 80k iterations with a learning rate of 0.01 and then for another 40k iterations with 0.001. For **setting-2** and **setting-3**, we employ the configurations of the Caffe2-Detectron (Girshick et al., 2018). We train all models on 8 GPUs and 2 images per GPU. Each image is re-scaled to its shorter side of 800 pixels. In particular, for setting-2, the learning rate (LR) is initialized as 0.02 and is decreased by a factor of 0.1 after 60k and 80k iterations and finally terminates at 90k iterations. This is referred as the 1x schedule in Detectron. In setting-3, the LR schedule is twice as long as the 1x schedule with the LR decay points scaled twofold proportionally, referred as 2x schedule. For all settings, we set weight decay to 0 for both $\gamma$ and $\beta$ following (Wu & He, 2018).

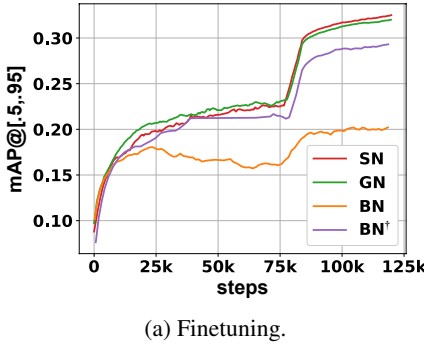 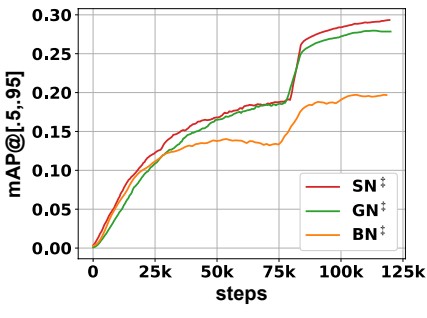

(a) Finetuning.  (b) Training from Scratch.

Figure 8: Average precision (AP) curves of Faster R-CNN on the *2017 val* set of COCO. (a) plots the results of finetuning pretrained networks. (b) shows training the models from scratch.

All the above models are trained in the *2017 train* set of COCO by using SGD with a momentum of 0.9 and a weight decay of $10^{-4}$ on the network parameters, and tested in the *2017 val* set. We report the standard metrics of COCO, including average precisions at IoU=0.5:0.05:0.75 (AP), IoU=0.5 ($AP_{.5}$), and IoU=0.75 ($AP_{.75}$) for both bounding box ($AP^b$) and segmentation mask ($AP^m$). Also, we report average precisions for small ($AP_s$), medium ($AP_m$), and large ($AP_l$) objects.

**Results of Setting-1.** As shown in Table 7, SN is compared with both BN and GN in the Faster R-CNN. In this setting, the layers up to conv4 of ResNet50 are used as *backbone* to extract features, and the layers of conv5 are used as the Region-of-Interest *head* for classification and regression. As the layers are inherited from the pretrained model, both the backbone and head involve normalization layers. Different results of Table 7 use different normalization methods in the backbone and head. Its upper part shows results of finetuning the ResNet50 models pretrained on ImageNet. The lower part compares training COCO from scratch without pretraining on ImageNet.

In the upper part of Table 7, the baseline is denoted as $BN^{\dagger}$, where the BN layers are frozen. We see that freezing BN performs significantly better than finetuning BN (29.6 *v.s.* 19.3). SN and GN enable finetuning the normalization layers, where SN obtains the best-performing AP of 33.0 in this setting. Fig.8 (a) compares their AP curves.

As reported in the lower part of Table 7, SN and GN allow us to train COCO from scratch without pretraining on ImageNet, and they still achieve competitive results. For instance, 29.5 of $SN^{\ddagger}$ outperforms $BN^{\ddagger}$ by a large margin of 9.5 AP and $GN^{\ddagger}$ by 1.2 AP. Their learning curves are compared in Fig.8 (b).

**Results of Setting-2 and -3.** The results of setting-2 and setting-3 are presented in the paper.

## E  SEMANTIC IMAGE PARSING

**Setting.** Similar to object detection, semantic image segmentation also benefits from large input size, making the minibatch size is small during training. We use 2 samples per GPU for ADE20K and 1 sample per GPU for Cityscapes. We employ the open-source software in PyTorch[2] and only replace the normalization layers in CNNs with the other settings fixed. For both datasets, we use DeepLab (Chen et al., 2018) with ResNet50 as the backbone network, where $output\_stride = 8$ and the last two blocks in the original ResNet contains atrous convolution with $rate = 2$ and $rate = 4$ respectively. Following (Zhao et al., 2017), we employ "poly" learning rate policy with $power = 0.9$ and use the auxiliary loss with the weight $0.4$ during training. The bilinear operation is adopted to upsmaple the score maps in the validation phase.

**ADE20K.** SyncBN and GN adopt the pretrained models on ImageNet. SyncBN collects the statistics from 8 GPUs. Thus the actual "batchsize" is 16 during training. To evaluate the performance of SN, we use SN $(8, 2)$ in ImageNet as the pretrained model. For all models, we resize each image to $450 \times 450$ and train for $100,000$ iterations. We performance multi-scale testing with $input\_size = \{300, 400, 500, 600\}$.

---

[2]https://github.com/CSAILVision/semantic-segmentation-pytorch

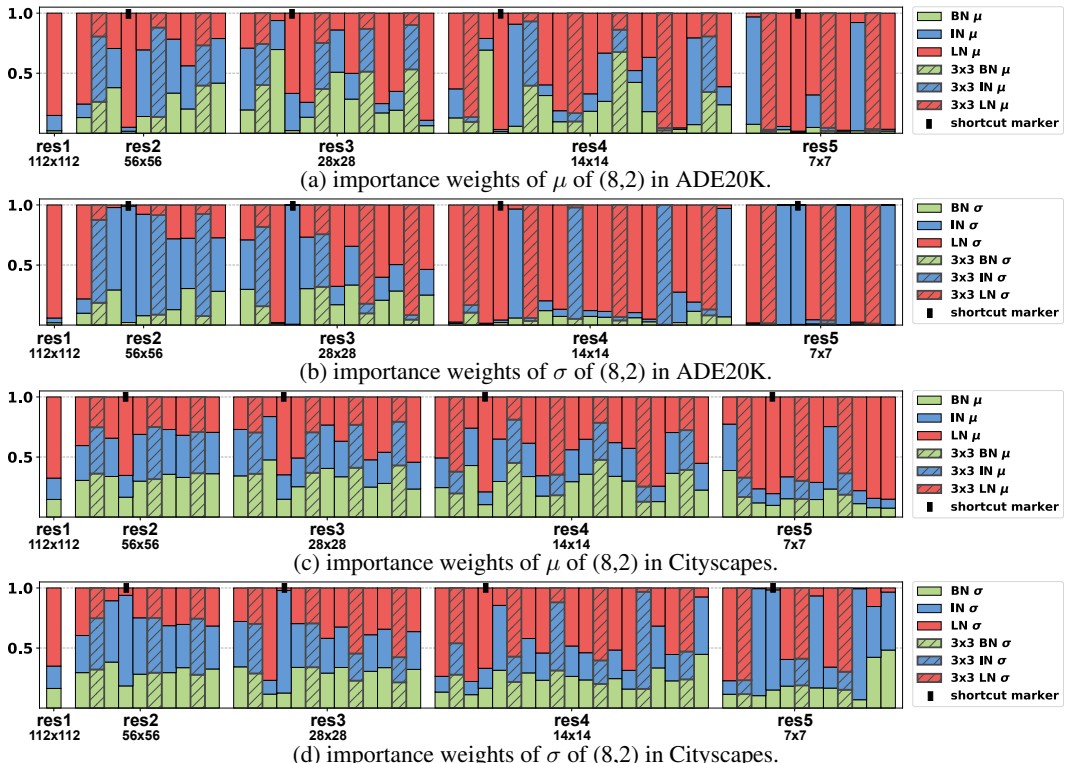

Figure 9: **Selected normalizers of each SN layer in ResNet50 for semantic image parsing in ADE20K and Cityscapes**. There are 53 SN layers. (a,b) show the importance weights for $\mu$ and $\sigma$ of $(8, 2)$ in ADE20K, while (c,d) show those of $(8, 2)$ in Cityscapes. The $y$-axis represents the importance weights that sum to 1, while the $x$-axis shows different residual blocks of ResNet50. The SN layers in different places are highlighted differently. For example, the SN layers follow the $3 \times 3$ conv layers are outlined by shaded color, those in the shortcuts are marked with '■', while those follow the $1 \times 1$ conv layers are in flat color.

**Cityscapes.** For all models, we finetune from their pretrained ResNet50 models. SN finetunes from $(8, 2)$. For all models, the batchsize is $8$ in finetuning. We use random crop with the size $713 \times 713$ and train for 700 epoches. For multi-scale testing, the inference scales are $\{1.0, 1.25, 1.5, 1.75\}$.

**Ablation Study.** Fig.9 compares the importance weights of SN in ResNet50 trained on both ADE20K and Cityscapes. We see that even when the models and tasks are the same, different training data encourage SN to choose different normalizers.

## F    ARTISTIC IMAGE STYLIZATION

We evaluate SN in the task of artistic image stylization. We adopt a recent advanced approach (Johnson et al., 2016), which jointly minimizes two loss functions. Specifically, one is a feature reconstruction loss that penalizes an output image when its content is deviated from a target image, and the other is a style reconstruction loss that penalizes differences in style (*e.g.* color, texture, exact boundary). Johnson et al. (2016); Huang & Belongie (2017) show that IN works better than BN in this task.

We compare SN with IN and BN using VGG16 (Simonyan & Zisserman, 2014) as backbone network. All models are trained on the COCO dataset (Lin et al., 2014). For each model in training, we resize each image to $256 \times 256$ and train for $40,000$ iterations with a batch size setting of $(1, 4)$. We do not employ weight decay or dropout. The other training protocols are the same as (Johnson et al., 2016). In test, we evaluate the trained models on $512 \times 512$ images selected following (Johnson et al., 2016).

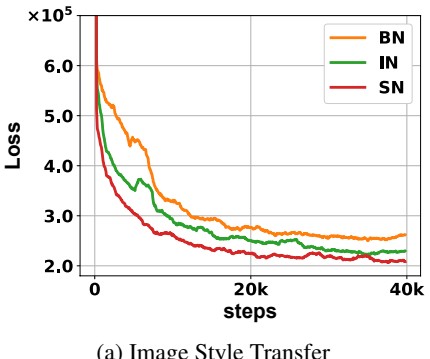 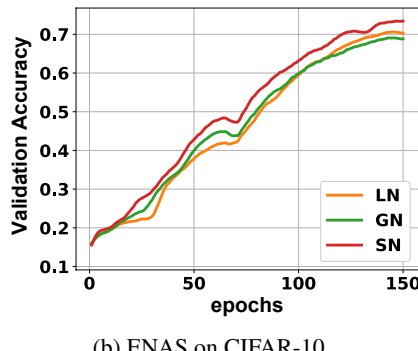

(a) Image Style Transfer (b) ENAS on CIFAR-10

Figure 10: (a) shows the losses of BN, IN, and SN in the task of image stylization. SN converges faster than IN and BN. As shown in Fig.1 and the supplementary material, SN adapts its importance weight to IN while producing comparable stylization results. (b) plots the accuracy on the validation set of CIFAR-10 when searching network architectures.

Fig.10 (a) compares the style and feature reconstruction losses. We see that SN enables faster convergence than both IN and BN. As shown in Fig.1 (a), SN automatically selects IN in image stylization. Some stylization results are visualized in Fig.11.

## G NEURAL ARCHITECTURE SEARCH

We investigate SN in LSTM for efficient neural architecture search (ENAS) (Pham et al., 2018), which is designed to search the structures of convolutional cells. In ENAS, a convolutional neural network (CNN) is constructed by stacking multiple convolutional cells. It consists of two steps, training controllers and training child models. A controller is a LSTM whose parameters are trained by using the REINFORCE (Williams, 1992) algorithm to sample a cell architecture, while a child model is a CNN that stacks many sampled cell architectures and its parameters are trained by back-propagation with SGD. In (Pham et al., 2018), the LSTM controller is learned to produce an architecture with high reward, which is the classification accuracy on the validation set of CIFAR-10 (Krizhevsky, 2009). Higher accuracy indicates the controller produces better architecture.

We compare SN with LN and GN by using them in the LSTM controller to improve architecture search. As BN is not applicable in LSTM and IN is equivalent to LN in fully-connected layer (*i.e.* both compute the statistics across neurons), SN combines LN and GN in this experiment. Fig.10 (b) shows the validation accuracy of CIFAR10. We see that SN obtains better accuracy than both LN and GN.

## H BACK-PROPAGATION OF SN

For the software without auto differentiation, we provide the backward computations of SN below. Let $\hat{h}$ be the output of the SN layer represented by a 4D tensor $(N, C, H, W)$ with index $n, c, i, j$. Let $\hat{h} = \gamma\tilde{h}+\beta$ and $\tilde{h} = \frac{h-\mu}{\sqrt{\sigma^2+\epsilon}}$, where $\mu = w_{\text{bn}}\mu_{\text{bn}}+w_{\text{in}}\mu_{\text{in}}+w_{\text{ln}}\mu_{\text{ln}}$, $\sigma^2 = w_{\text{bn}}\sigma^2_{\text{bn}}+w_{\text{in}}\sigma^2_{\text{in}}+w_{\text{ln}}\sigma^2_{\text{ln}}$, and $w_{\text{bn}} + w_{\text{in}} + w_{\text{ln}} = 1$. Note that the importance weights are shared among the means and variances for clarity of notations. Suppose that each one of $\{\mu, \mu_{\text{bn}}, \mu_{\text{in}}, \mu_{\text{ln}}, \sigma^2, \sigma^2_{\text{bn}}, \sigma^2_{\text{in}}, \sigma^2_{\text{ln}}\}$ is reshaped into a vector of $N \times C$ entries, which are the same as the dimension of IN's statistics. Let $\mathcal{L}$ be the loss function and $(\frac{\partial\mathcal{L}}{\partial\mu})_n$ be the gradient with respect to the $n$-th entry of $\mu$.

We have

$$\frac{\partial\mathcal{L}}{\partial\tilde{h}_{ncij}} = \frac{\partial\mathcal{L}}{\partial\hat{h}_{ncij}} \cdot \gamma_c, \tag{6}$$

$$\frac{\partial\mathcal{L}}{\partial\sigma^2} = -\frac{1}{2(\sigma^2+\epsilon)} \sum_{i,j}^{H,W} \frac{\partial\mathcal{L}}{\partial\tilde{h}_{ncij}} \cdot \tilde{h}_{ncij}, \tag{7}$$

$$\frac{\partial \mathcal{L}}{\partial \mu} = -\frac{1}{\sqrt{\sigma^2 + \epsilon}} \sum_{i,j}^{H,W} \frac{\partial \mathcal{L}}{\partial \tilde{h}_{ncij}}, \tag{8}$$

$$\frac{\partial \mathcal{L}}{\partial h_{ncij}} = \frac{\partial \mathcal{L}}{\partial \tilde{h}_{ncij}} \cdot \frac{1}{\sqrt{\sigma^2 + \epsilon}} + \Big[\frac{2w_{\text{in}}(h_{ncij} - \mu_{\text{in}})}{HW} \frac{\partial \mathcal{L}}{\partial \sigma^2}$$

$$+ \frac{2w_{\text{ln}}(h_{ncij} - \mu_{\text{ln}})}{CHW} \sum_{c=1}^{C} (\frac{\partial \mathcal{L}}{\partial \sigma^2})_c + \frac{2w_{\text{bn}}(h_{ncij} - \mu_{\text{bn}})}{NHW} \sum_{n=1}^{N} (\frac{\partial \mathcal{L}}{\partial \sigma^2})_n\Big]$$

$$+ \Big[\frac{w_{\text{in}}}{HW} \frac{\partial \mathcal{L}}{\partial \mu} + \frac{w_{\text{ln}}}{CHW} \sum_{c=1}^{C} (\frac{\partial \mathcal{L}}{\partial \mu})_c + \frac{w_{\text{bn}}}{NHW} \sum_{n=1}^{N} (\frac{\partial \mathcal{L}}{\partial \mu})_n\Big], \tag{9}$$

The gradients for $\gamma$ and $\beta$ are

$$\frac{\partial \mathcal{L}}{\partial \gamma} = \sum_{n,i,j}^{N,H,W} \frac{\partial \mathcal{L}}{\partial \hat{h}_{ncij}} \cdot \tilde{h}_{ncij}, \tag{10}$$

$$\frac{\partial \mathcal{L}}{\partial \beta} = \sum_{n,i,j}^{N,H,W} \frac{\partial \mathcal{L}}{\partial \hat{h}_{ncij}}, \tag{11}$$

and the gradients for $\lambda_{\text{in}}, \lambda_{\text{ln}}$, and $\lambda_{\text{bn}}$ are

$$\frac{\partial \mathcal{L}}{\partial \lambda_{\text{in}}} = w_{\text{in}}(1 - w_{\text{in}}) \sum_{n,c}^{N,C} ((\frac{\partial \mathcal{L}}{\partial \mu})_{nc} \mu_{\text{in}} + (\frac{\partial \mathcal{L}}{\partial \sigma^2})_{nc} \sigma_{\text{in}}^2)$$

$$- w_{\text{in}} w_{\text{ln}} \sum_{n,c}^{N,C} ((\frac{\partial \mathcal{L}}{\partial \mu})_{nc} \mu_{\text{ln}} + (\frac{\partial \mathcal{L}}{\partial \sigma^2})_{nc} \sigma_{\text{ln}}^2)$$

$$- w_{\text{in}} w_{\text{bn}} \sum_{n,c}^{N,C} ((\frac{\partial \mathcal{L}}{\partial \mu})_{nc} \mu_{\text{bn}} + (\frac{\partial \mathcal{L}}{\partial \sigma^2})_{nc} \sigma_{\text{bn}}^2), \tag{12}$$

$$\frac{\partial \mathcal{L}}{\partial \lambda_{\text{ln}}} = w_{\text{ln}}(1 - w_{\text{ln}}) \sum_{n,c}^{N,C} ((\frac{\partial \mathcal{L}}{\partial \mu})_{nc} \mu_{\text{ln}} + (\frac{\partial \mathcal{L}}{\partial \sigma^2})_{nc} \sigma_{\text{ln}}^2)$$

$$- w_{\text{in}} w_{\text{ln}} \sum_{n,c}^{N,C} ((\frac{\partial \mathcal{L}}{\partial \mu})_{nc} \mu_{\text{in}} + (\frac{\partial \mathcal{L}}{\partial \sigma^2})_{nc} \sigma_{\text{in}}^2)$$

$$- w_{\text{ln}} w_{\text{bn}} \sum_{n,c}^{N,C} ((\frac{\partial \mathcal{L}}{\partial \mu})_{nc} \mu_{\text{bn}} + (\frac{\partial \mathcal{L}}{\partial \sigma^2})_{nc} \sigma_{\text{bn}}^2), \tag{13}$$

$$\frac{\partial \mathcal{L}}{\partial \lambda_{\text{bn}}} = w_{\text{bn}}(1 - w_{\text{bn}}) \sum_{n,c}^{N,C} ((\frac{\partial \mathcal{L}}{\partial \mu})_{nc} \mu_{\text{bn}} + (\frac{\partial \mathcal{L}}{\partial \sigma^2})_{nc} \sigma_{\text{bn}}^2)$$

$$- w_{\text{in}} w_{\text{bn}} \sum_{n,c}^{N,C} ((\frac{\partial \mathcal{L}}{\partial \mu})_{nc} \mu_{\text{in}} + (\frac{\partial \mathcal{L}}{\partial \sigma^2})_{nc} \sigma_{\text{in}}^2)$$

$$- w_{\text{ln}} w_{\text{bn}} \sum_{n,c}^{N,C} ((\frac{\partial \mathcal{L}}{\partial \mu})_{nc} \mu_{\text{ln}} + (\frac{\partial \mathcal{L}}{\partial \sigma^2})_{nc} \sigma_{\text{ln}}^2). \tag{14}$$

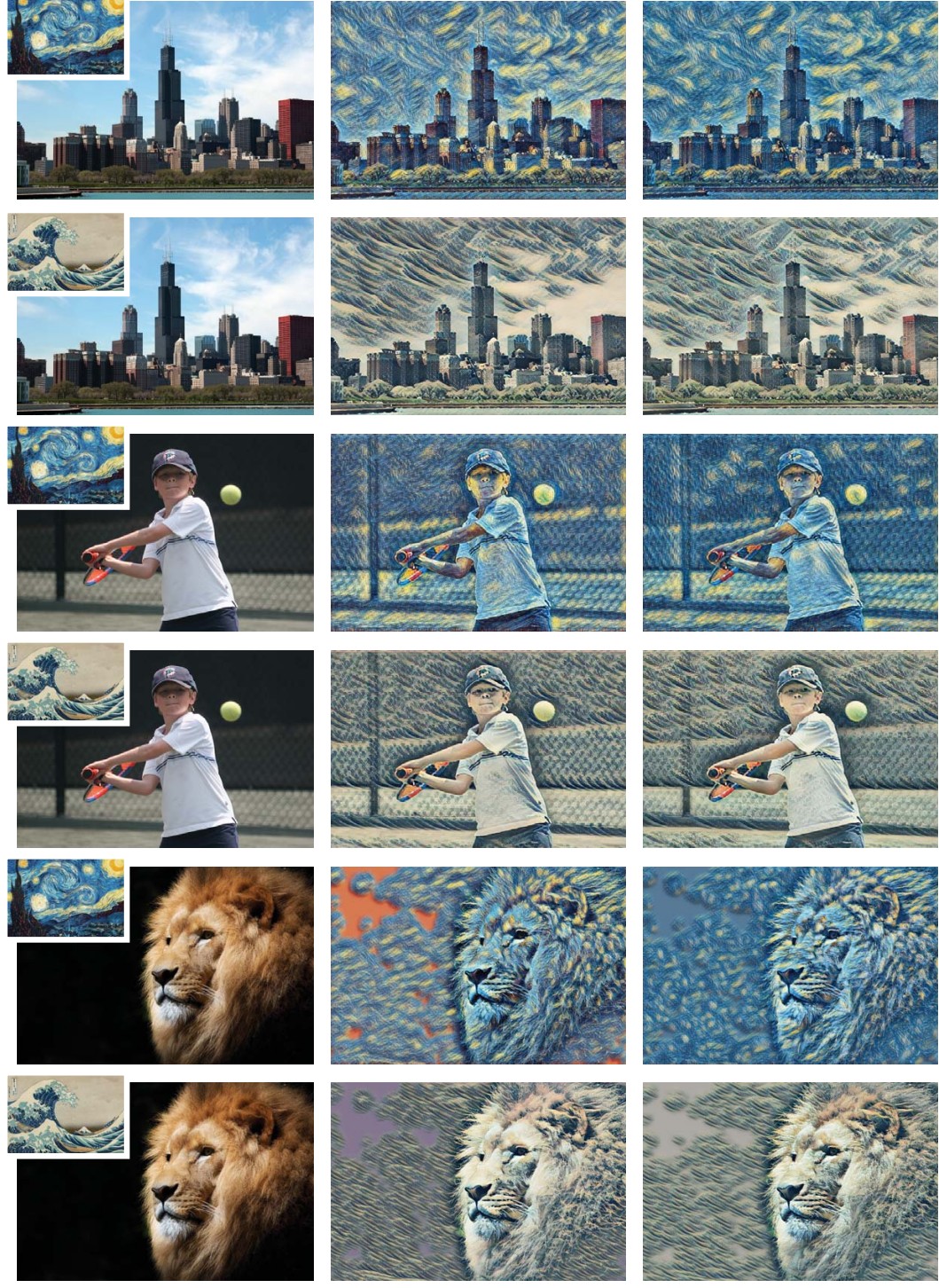

Figure 11: **Results of Image Stylization.** The first column visualizes the content and the style images. The second and third columns are the results of IN and SN respectively. SN works comparably well with IN in this task.

