# OpenReview forum: "Differentiable Learning-to-Normalize via Switchable Normalization"
_ICLR.cc/2019/Conference_

### Official Review · AnonReviewer3 · 2018-11-01
**Neat motivation and very extensive experiments**

**Rating:** 7
**Confidence:** 5

**Review:**

In this work, the authors propose Switchable Normalization (SN), which *learns* to switch / select different normalization algorithms (including batch normalization (BN), Instance Normalization (IN), Layer Normalization (LN)) in layers of the networks and in different applications. The idea is motivated by observations (shown in Fig 1) that, 1) different tasks tend to have applied different normalization methods; 2) some normalization methods (e.g. BN) are fragile to very small batch size.

The authors propose a general form for different normalization methods, which is a Gaussian normalization and then scale and shift by scalars. Different normalization methods utilize different statistics as the mean and the variance of the Gaussian normalization. The authors further propose to learn the combination weights on mean and variance, which is w_k and w'_k in Eqn (3). To avoid duplicate computation, the authors also do some careful simplification on computing mean and variance with all of the three normalization methods.

In the experiment part, the authors demonstrate the effectiveness of the proposed SN method on various kinds of tasks, including ImageNet classification, object detection and instance segmentation in COCO, semantic image parsing and video recognition. In all of the tasks tested, which also cover the common application in computer vision, SN shows superior and robust performance.

Pros:
+ Neat motivation;
+ Extensive experiments;
+ Clear illustration;

Cons
- There are still some experiment results missing, as the authors themselves mentioned in the Kinetics section (but the reviewer thinks it would be ready);
- In Page 3 the training section and Page 4, the first paragraph, it mentioned Θ and Φ (which are the weights for different normalization methods) are jointly trained and different from the previous iterative meta-learning style method. The authors attribute "In contrast, SN essentially prevents overfitting by choosing normalizers to improve both learning and generalization ability as discussed below". The reviewer does not see it is well justified and the reviewer thinks optimizing them jointly could lead to instability in the training (but it did not happen in the experiments). The authors should justify the jointly training part better.
- Page 5 the final paragraph, the reviewer does not see the point there. "We would also like to acknowledge the contributions of previous work that explored spatial region (Ren et al., 2016) and conditional normalization (Perez et al., 2017). "  Please make it a bit more clear how these works are related.

---

> ### Author Response · Authors · 2018-11-19
> **Responses to AnonReviewer3**
>
> Thanks for your reviews. We would like to provide feedback as below.
>
> 1. In the Kinetics section, SN achieves top1/top5 accuracy of 73.5/91.2 that outperforms BN and GN by 0.2/0.5 and 0.5/0.6 respectively.
>
> 2. We will add more discussions of the joint training process. In fact, SN optimizes both network parameters (\Theta) and control parameters (\Phi) in the same training set, because \Phi in SN controls the ratios between different normalizers where each one has certain regularization capacity. Their combination provides stronger regularization to prevent over-fitting in training.
>
> 3. In the related work section, we would like to acknowledge more normalization methods in the literature. We will discuss more details of them.
>
> Besides the above feedback, we would also like to highlight that SN would inspire future work of both applications and theories, as discussed in the "response to AnonReviewer4" in https://openreview.net/forum?id=ryggIs0cYQ&noteId=B1x5xwU-a7&noteId=B1x5xwU-a7

---

### Official Review · AnonReviewer4 · 2018-11-07
**A clear paper with clear contributions**

**Rating:** 7
**Confidence:** 4

**Review:**

Summary:
(1) This paper proposes the concept of Switchable Normalization (SN), which learns a weighted combination of three popular/existing normalization techniques, Instance Normalization (IN) for channel-wise, Layer Normalization (LN) for layer-wise, and Batch Normalization (BN) for minibatch-wise.
(2) Some interesting technical details: a) A softmax is learned to automatically determine the importance of each normalization; b) Reuse of the computation to accelerate.  c) Geometric view of different normalization methods.
(3) Extensive experimental results to show the performance improvement. Investigation on the learned importance on different parts of networks and tasks.


Comments:

The writing of this paper is excellent, and contributions are well presented and demonstrated.
It is good for the community to know SN is an option to consider. Therefore, I vote to accept the paper.

However, the proposed method itself is not significant, given many existing efforts/algorithms; it is almost straightforward to do so, without any challenges.

Here is a more challenging question for the authors to consider: Given the general formulation of normalization methods in (2), it sees more interesting to directly learn the pixel set I_k. The proposed SN can be considered as a weak version to learn the pixel set: SN employs the three candidates set pre-defined by the existing methods, and learns a weighted combination over the  “template” sets. This is easy to do in practice, and it has shown promising results. A natural idea to learn more flexible pixel set, and see the advantages. Any thoughts?

---

> ### Author Response · Authors · 2018-11-08
> **Responds to AnonReviewer4**
>
> Dear AnonReviewer4,
>
> The authors appreciate you're voting to accept and you're definitely a responsible reviewer. While we still have two missing reviews from R1 and R2.
>
> Thanks for your good question for us to consider. In fact, the authors have been working on it as SNv2, which will be released very soon. However, the details of SNv2 might be out of the scope of this paper.
>
> The authors would like to respond to your comment "the proposed method itself is not significant, given many existing efforts/algorithms; it is almost straightforward to do so, without any challenges". We believe you're saying that the technical challenge of SN is not significant because SN combines existing normalizers in a straightforward way.
>
> To be honest, the authors might not agree with the above comment. (1) SN is not only widely applicable, but also a thought provoking method because it's the first time to demonstrate different convolutional layers should have different normalizers. This viewpoint may undoubtedly inspire many areas in deep learning as pointed out in the paper.
>
> (2) The authors would also highlight that SN is designed to be as concise and effective as possible without any hyper-parameter and easy to implement and use. We believe that the current formulation of SN is important and valuable, and it should be presented to the community.
>
> Although SN has a straightforward form that uses softmax to combine normalizers, its theoretical analysis (such as learning dynamics and generalization) is already a big challenge because theory of BN, IN, and LN should be built up before analyzing SN.
>
> While analyzing any one of these normalizers is still an open problem, not to mention to investigate the interactions between them, despite their interactions are captured by using a "simple" softmax function.
>
> We also might not agree that the papers submitted to this top-tier venue would be "easy to do in practice" especially for those demonstrated new perspective and inspiration for deep learning, otherwise they would be already present in any where else.
>
> Frankly, the authors believe the SN paper deserves a better score. Do the above feedback persuade you to raise your rating?

---

### Official Review · AnonReviewer1 · 2018-11-11
**Simple approach that works**

**Rating:** 7
**Confidence:** 3

**Review:**

This paper considers normalization by learning to average across well-known normalization techniques.
This meta-procedure almost by definition can yield better results, and this is demonstrated nicely by the authors.
The idea is simple and easy to implement, and easy works in this case.

I would like to hear more about the connections to other meta-learning procedures, by expanding the discussion on page 3.
That discussion is very interesting but quite short, and I am afraid I can't see how SN avoids overfitting compared to other approaches.
Also, the section on Inference in page 4 is unclear to me. What parameters are being inferred and why is the discussion focused on convergence instead?

---

### Public Comment · (anonymous) · 2018-12-20
**interesting idea. Could you provide some results on CIFAR-10/100?**

the idea to uniform some of the best normalization techniques is quite interesting.
Quick question: does it make sense to use separate control parameters for the means and stds?
additional experiment request: Could you please provide us with a benchmark on CIFAR-10/100, which is less resource demanding so that people with limited computational resources can also have a quick look at the efficacy of the method? thanks

---

### Meta-Review · Area_Chair1 · 2018-12-14
**Well motivated simple idea and solution that work well in practice**

**Confidence:** 5
**Recommendation:** Accept (Poster)

**Metareview:**

This paper proposes Switchable Normalization (SN) that leans how to combine three existing normalization techniques for improved performance. There is a general consensus that that the paper has good quality and clarity, is well motivated, is sufficiently novel, makes clear contributions for training deep neural networks, and provides convincing experimental results to show the advantages of the proposed SN.